# Rare earth stibolyl and bismolyl sandwich complexes

Noah Schwarz[1], Florian Bruder[2], Valentin Bayer[3,4], Eufemio Moreno-Pineda [5,6,7], Sebastian Gillhuber [1], Xiaofei Sun [1], Joris van Slageren [3] ✉, Florian Weigend [2] ✉ & Peter W. Roesky [1,8] ✉

The design of molecular rare earth complexes to achieve unique magnetic and bonding properties is a growing area of research with possible applications in advanced materials and molecular magnetics. Recent efforts focus on developing ligand frameworks that can enhance magnetic characteristics. Here we show the synthesis and characterization of a class of rare earth complexes, $[(\eta^5\text{-}C_4R_4Sb)Ln(\eta^8\text{-}C_8H_8)]$ and $[(\eta^5\text{-}C_4R_4Bi)Ln(\eta^8\text{-}C_8H_8)]$, featuring $\eta^5$-coordinated stibolyl and bismolyl ligands. The ligand aromaticity and bonding situation within these complexes are investigated by quantum chemical calculations. Magnetic studies of the $Er^{III}$ analogues reveal large barriers and intriguing properties, including waist-restricted hysteresis and slow relaxation of the magnetization, making them single-molecule magnets. Comparison between the experimental barrier and CASSCF-SO calculations indicates that relaxation in all systems occurs through high-energy excited states. These findings suggest that stibolyl and bismolyl ligands can be promising candidates for achieving high-energy barriers in Er-based SMMs, offering a pathway to molecular designs with enhanced magnetic properties.

Sandwich complexes have long been a captivating class of compounds, with the discovery of ferrocene marking a significant milestone[1]. Since then, cyclopentadienyl ($Cp^- = C_5H_5^-$) ligands have been widely employed and remain the dominating ligand system in organometallic rare earth chemistry. Their electron-donating ability significantly influences the structure and reactivity of rare earth complexes, enabling the fine-tuning of these compounds for various applications including catalysis, luminescent materials, and single molecule magnets (SMMs). While Cp ligands have played a prominent role, exploration beyond them has led to the utilization of other carbon-based ring systems for rare earth coordination.

The COT dianion ($C_8H_8^{2-}$), alongside the extensively studied cyclopentadienyl ligand and its derivatives, stands out as one of the most essential ligands in organo-f-element chemistry[2]. The sterically demanding, planar dianion represents a valuable alternative to the smaller cyclopentadienyl ligands[3,4]. Among the longest-known rare earth COT complexes are symmetrical sandwich complexes containing $[Ln(COT)_2]^-$ anions[5,6]. In 1993, silyl-substituted cyclooctatetraenyl ligands were introduced by Cloke and coworkers[7]. Building upon this, in 1998 Edelmann achieved the synthesis of homoleptic rare earth triple-decker complexes, denoted as $[Ln^{III}_2(\eta^8\text{-COT''})_3]$ (Ln = Ce, Nd, Sm, COT'' = $1,4\text{-}(Me_3Si)_2C_8H_6^{2-})$[8]. Recently, our group

[1]Institute of Inorganic Chemistry, Karlsruhe Institute of Technology Kaiserstrasse 12, Karlsruhe, Germany. [2]Fachbereich Chemie, Philipps-Universität Marburg Hans-Meerwein-Straße 4, Marburg, Germany. [3]Institute of Physical Chemistry, University of Stuttgart Pfaffenwaldring 55, Stuttgart, Germany. [4]Institute of Inorganic Chemistry, University of Stuttgart Pfaffenwaldring 55, Stuttgart, Germany. [5]Universidad de Panamá, Facultad de Ciencias Naturales, Exactas y Tecnología, Depto. de Química-Física, Panamá, Panamá. [6]Universidad de Panamá, Facultad de Ciencias Naturales, Exactas y Tecnología, Grupo de Investigación de Materiales, Panamá, Panamá. [7]Physikalisches Institut, Karlsruhe Institute of Technology Kaiserstrasse 12, Karlsruhe, Germany. [8]Institute of Nanotechnology, Karlsruhe Institute of Technology Kaiserstrasse 12, Karlsruhe, Germany. ✉e-mail: slageren@ipc.uni-stuttgart.de; florian.weigend@chemie.uni-marburg.de; roesky@kit.edu

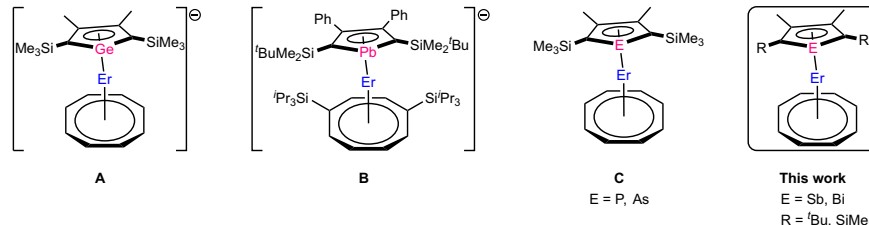

**Fig. 1 | Previous work.** Molecular structures of [Er(η⁵-LᴳᵉGe)(η⁸-COT)]⁻ (**A**)[21], [Er(η⁵-Lᴾᵇ)(η⁸-COTᵀᴵᴾˢ)]⁻ (**B**)[22], and [Er(η⁵-Dtp/Dsas)(η⁸-COT) **C**[24,25]. The cations of (**A**, **B**) are omitted for clarity. (Lᴳᵉ = 3,4-dimethyl-2,5-bis(trimethylsilyl)germolyl, Lᴾᵇ = 3,4- diphenyl-2,5-bis(*tert*-butyldimethylsilyl)plumbolyl, COTᵀᴵᴾˢ = 1,4-(ⁱPr₃Si)₂C₈H₆²⁻, Dsp = 3,4-dimethyl-2,5-bis(trimethylsilyl)phospholyl, Dsas = 3,4-dimethyl-2,5-bis(trimethylsilyl)arsolyl).

obtained a homoleptic mixed valent rare earth quadruple-decker complex [Smᴵᴵᴵ/ᴵᴵ/ᴵᴵᴵ₃(η⁸-COTᵀᴵᴾˢ)₄](COTᵀᴵᴾˢ = 1,4-(ⁱPr₃Si)₂C₈H₆²⁻)[9] and the cyclocenes [*cyclo*-Mᴵᴵ(μ-η⁸:η⁸-COTᵀᴵᴾˢ)]₁₈ (M = Sr, Sm, Eu)[10].

Also, heterocyclic ring systems, with at least one ring carbon atom replaced by a heteroatom, have emerged as a prominent ligand class in recent years. The utilization of dianionic group 13 metalloles, starting with boroles, for the coordination of iron dates back as early as 1986[11]. In rare earth chemistry, borolide ligands have been employed in the design of SMMs[12–15]

The chemistry of group 14 metalloles has been well described in the literature[16]. In addition to their dianionic structure, monoanions are also accessible through functionalization of the Si or Ge atom[17]. Recently, both silole and germole dianions have made their way into rare earth coordination chemistry. Silole COT complexes of La and Er were reported by Layfield and our group[18,19]. We also disclosed polymeric germole COT complexes of La and Ce[19]. Other germole complexes such as [Cp*(η⁵-Lᴳᵉ)Y]₂ (Cp* = C₅Me₅⁻, Lᴳᵉ = 3,4-dimethyl-2,5-bis(trimethylsilyl)germolyl) and [K(2.2.2-crypt)][(η⁵-Lᴳᵉ)Er(η⁸-COT)] (**A**, Fig. 1) have also been synthesized[20,21]. In 2022, we successfully incorporated plumbole, the heaviest group 14 metallole, into a series of [LnCOTᵀᴵᴾˢ]⁺ frameworks and the Er congener [(η⁵-Lᴾᵇ)Er(η⁸-COTᵀᴵᴾˢ)]⁻ (Lᴾᵇ = 3,4-diphenyl-2,5-bis(*tert*-butyldimethylsilyl)plumbolyl) (**B**, Fig. 1) displays magnetic hysteresis up to 5 K[22].

The utilization of group 15 phospholyl and arsolyl ligands has a long history, particularly in the case of phospholes, with a significant number of complexes reported in the literature[23]. Gao successfully synthesized the COT-based erbium-phospholyl complex [(η⁵-Dsp)Er(η⁸-COT)] (Dsp = [3,4-dimethyl-2,5-bis(trimethylsilyl)phospholyl]⁻) (**C**, Fig. 1). This complex exhibited magnetic hysteresis up to 9 K[24]. Recently, we also succeeded in synthesizing the isostructural arsolyl complexes[25]. In 2019, the group of Mills discovered a dysprosium-based SMM, featuring the cationic fragment [(η⁵-Dtp)₂Dy]⁺ (Dtp = 3,4-diphenyl-2,5-bis(*tert*-butyl)phospholyl), which demonstrated hysteresis up to 48 K, a remarkable achievement comparable to Cp-based systems that have dominated the field[26]. Since then, another phospholyl based dysprosium system with SMM performance very close to Cp based systems was published[27].

While phospholyl and arsolyl ligands have received considerable attention, their heavier congeners, stibolyls and bismolyls, have not been extensively studied. Although ferrocene derivatives with such heterocyclic ligands were initially synthesized in the early 1990s, showcasing intriguing secondary bonding interactions between the heavy atoms[28–30], the only other reported use of these ligands has been focused on the synthesis of alkali metal complexes[31,32]. While setting up this manuscript, Demir published a η¹-coordinated bismole complex [(C₅Me₄H)Y(μ-η¹-(BiC₄Me₄)]₂, in which the Bi atom bridges two Y atoms via σ-bonds. However, no π-coordination was observed[33]. The lack of π-coordinated heavier group 15 heterocyclopentadienyls in f-element coordination chemistry is surprising, given that their lighter congeners have been known and used for such a long time.

Herein, we showcase sandwich complexes of the rare earth elements featuring π-coordinated stibolyl and bismolyl ligands. Besides the synthetic challenge, we were aiming to design SMMs by employing these underexplored ligand systems and investigate the influence of heavier atoms on their magnetic properties. Additionally, the newly synthesized compounds were analyzed by quantum chemical calculations with TURBOMOLE[34,35] at the def2-TZVP[36,37]/PBE0[38,39] level.

## Results and discussion

Initially, we aimed for the neutral chloro-stiboles and -bismoles by the reaction of the chlorides SbCl₃ and BiCl₃ with substituted zirconacycles[40]. However, these reactions were limited in scope, with the most prominent challenge being the thermal lability of the Bi-Cl bond, leading to decomposition at room temperature. For this reason, we changed to an alternative synthetic pathway starting from 1,4-dilithiobutadienes[41]. Reaction of the latter with PhECl₂ (E = Sb, Bi) gave phenyl-stiboles and -bismoles, that could subsequently be reduced by the addition of an excess amount of potassium in THF (Fig. 2).

By subsequent filtration and washing with *n*-pentane, we successfully isolated **1-Sb**, **2-Sb** and **1-Bi** in analytically pure form in yields of 76%, 33% and 59%, respectively. All potassium salts were characterized by NMR spectroscopy (Fig. S1–S6, Supplementary Methods). Comparison of the obtained spectra to those of analogous phospholyl and arsolyl complexes confirmed the successful formation of the intended products. Due to difficulties in preparation of the ligand and the corresponding metal complexes, we did not synthesize the Bi analogue of **2-Sb**.

Having these precursors at hand, we proceeded with the reactions of the two potassium stibolyl complexes with [LnI(η⁸-C₈H₈)(thf)₂] (Ln = Y, Er, Tb) in *n*-pentane (Fig. 3a)[42]. Despite the poor solubility of the starting materials in this solvent, a rapid color change from light brown to yellow (Y) or orange (Er, Tb) and the precipitation of a colorless solid were observed within a few minutes. After filtration, the crystalline products could be obtained after concentrating the solutions and leaving them for several hours.

The molecular structures in the solid-state of **3-Ln** and **4-Ln** (Ln = Y, Er, Tb) were determined by single crystal X-ray diffraction (SCXRD) and the molecular structures of the Y and Er complexes are displayed in Fig. 3b. Notably, these compounds represent examples of f-element complexes with a η⁵-coordinating stibolyl ligand, as only complexes with iron and alkali metals are known so far[28,29,31,32,43,44].

All six complexes reveal monomeric structures with the rare earth ion being η⁵-coordinated by the stibolyl ligand on one side and η⁸-coordinated by the COT ligand on the other side (Fig. 3b). The Y-Sb bond lengths in **3-Y** and **4-Y** are 3.1141(8) Å and 3.0795(7) Å, respectively, which closely aligns with the only known examples of Y-Sb bonds reported in the literature[45,46]. In comparison, the previously published phospholyl complex [(η⁵-Dsp)Y(η⁸-COT)] features a Y-P bond length of 2.8261(6) Å, highlighting the larger steric demand of the antimony-based ligand and the larger ionic radius of Sb[24]. This disparity is also evident when examining the Y-Ct_Sb (Ct = centroid) distances for **3-Y** and **4-Y**, which are 2.408(3) Å and 2.376(2) Å, respectively, compared to 2.3545(8) Å in [(η⁵-Dsp)Y(η⁸-COT)]. The Y-Ct_COT distances, 1.731(2) Å for **3-Y** and 1.714(2) Å for **4-Y**, are similar to those observed in the phospholyl

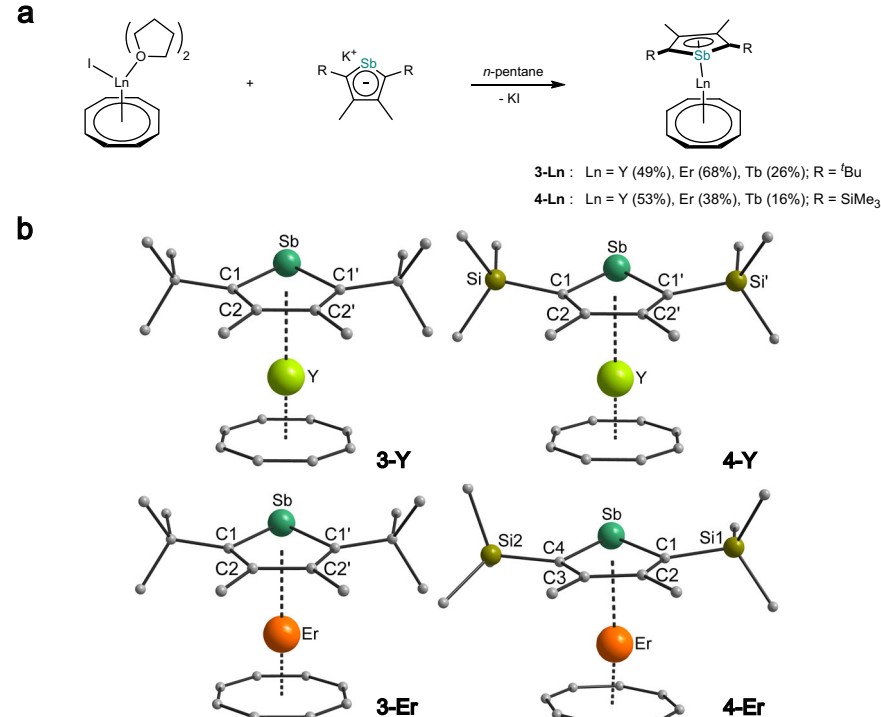

**Fig. 2 | Ligand Synthesis.** Synthesis of the potassium stibolyl (**1-Sb, 2-Sb**) and bismolyl (**1-Bi**) complexes[42].

**3-Ln** : Ln = Y (49%), Er (68%), Tb (26%); R = $^t$Bu
**4-Ln** : Ln = Y (53%), Er (38%), Tb (16%); R = SiMe$_3$

**Fig. 3 | Synthesis and Structures of rare earth stibolyl complexes. a** Synthesis of yttrium (**3-Y, 4-Y**), terbium (**3-Tb, 4-Tb**) and erbium (**3-Er, 4-Er**) stibolyl complexes. **b** Molecular structures of compounds **3-Y** (top left), **4-Y** (top right), **3-Er** (bottom left) and **4-Er** (bottom right) in the solid state. For better clarity hydrogen atoms are omitted and only one molecule of the asymmetric unit of **4-Er** is depicted[42].

complex (1.707(7) Å and 1.734(3) Å)[24]. The Ct$_{COT}$-Y-Ct$_{Sb}$ bond angles are 169.38(10)° and 168.92(3)° for **3-Y** and **4-Y**, respectively. Comparing the two yttrium structures, it is evident that replacing the $^t$Bu group with a SiMe$_3$ group does not result in significant structural differences.

Due to the slightly smaller size of erbium compared to yttrium[47], the Er-Sb bond length in **3-Er** is slightly shorter with 3.0883(4) Å, which can also be seen in the Er-Ct$_{Sb}$ distance with 2.385(2) Å, while the Ct$_{COT}$-Er-Ct$_{Sb}$ angle is almost identical. Since **4-Er** crystallized with different independent molecules in the unit cell, no detailed comparison to **4-Y** will be made here (for the structural parameters of complex **4-Er**, see Table S2 in the Supplementary Information). In all complexes, the stibolyl ring is notably tilted due to the steric demands of the antimony atom and the $^t$Bu groups, resulting in Er-C1 and Er-C2 bond lengths in **3-Er** of 2.757(3) and 2.650(3) Å, respectively. This tilting is absent in [Er(COT)Cp*] (Er-C distance of 2.575(3)-2.579(3) Å)[48]. However, in the more sterically demanding [Er(COT)Cp$^{ttt}$] (Cp$^{ttt}$ = C$_5$$^t$Bu$_3$H$_2$), the Er-C distances range from 2.580(2) to 2.649(2) Å, demonstrating a slight tilt of the Cp ligand, though not as pronounced as in our stibolyl complexes[49].

The $^1$H NMR spectrum of **3-Y** exhibits three distinctive signals: the COT protons are observed at δ = 6.35 ppm, the methyl groups resonate at δ = 1.94 ppm, and the $^t$Bu groups display a signal at δ = 1.30 ppm. Noteworthy, in the $^{13}$C{$^1$H} NMR spectrum, coupling between $^{89}$Y

and $^{13}$C atoms is evident. The signal of the α-carbons adjacent to the antimony is split into a doublet at δ = 187.5 ppm ($^1J_{C,Y}$ = 2.2 Hz). Furthermore, the carbon signals from the COT ring show a doublet at δ = 94.5 ppm ($^1J_{C,Y}$ = 3.0 Hz).

The $^1$H NMR spectrum of **4-Y** closely resembles that of **3-Y**, with no significant deviations, except for the signal of the SiMe$_3$ group appearing at lower frequencies (δ = 0.32 ppm). Analogously, the signal of the α-carbons of the stibolyl ring appears as a doublet at δ = 174.5 ppm with a coupling constant of $^1J_{C,Y}$ = 2.5 Hz. Notably, this signal is shifted towards lower frequencies compared to **3-Y** due to effects of the SiMe$_3$ group. Furthermore, the signal of the COT carbons splits into a doublet at δ = 94.3 ppm, with a coupling constant of $^1J_{C,Y}$ = 3.2 Hz.

The bonding situation within complexes **3-Y**, **3-Er**, **4-Y**, and **4-Er** is reminiscent of that of rare earth plumbole complexes previously reported by our group[22]. Mulliken overlap populations[50] (Supplementary Information, Table S5) between Sb and the rare earth ions amount to approximately 0.3 electrons (roughly half of the overlap observed for a typical single bond, e.g., 0.6 for SbH$_3$ or 0.7 for YH$_3$), indicating the presence of a weak covalent interaction of the stibolyl Sb atom with the rare earth metal. A considerable amount of this interaction can be attributed to the highest occupied molecular orbital (HOMO) and a second energetically close-lying occupied molecular

orbital (MO) of the complexes, which are depicted in Fig. 4a. For all four complexes, the HOMO exhibits a π-bonding interaction between one of the π orbitals of the stibolyl ligand with a 4d (Y) or 5d (Er) orbital of the rare earth element. Additional contributions arise from

interactions with the COT π system, which are more pronounced in the contour plots of **4-Y** and **4-Er**. The second MO describes the δ-bonding interaction of one of the COT π orbitals with another 4d (Y) or 5d (Er) orbital of the rare earth metal. For all complexes, this MO features additional mixing with a 5p orbital of the Sb atom. Confirming Mulliken population analyses are provided in the Supplementary Notes (Tables S6 and S7), which also indicate that the f orbitals of Er do not contribute significantly to the orbitals relevant to the rare earth-Sb bonding.

After successfully establishing these stibolyl ligands in complexes **3** and **4**, we were interested to see whether it is possible to form similar complexes with the heavier bismolyl anion. The use of Bi-based ligands is intriguing due to bismuth's strong spin-orbit coupling, which upon coordination to a metal center, could significantly influence the magnetic properties of the complex. Moreover, in comparison to the other p-block metals of the 6th period (Tl, Pb, Po), bismuth shows significantly lower toxicity, further enhancing its appeal as a potential ligand in coordination chemistry. Therefore, reactions between the potassium bismolyl **1-Bi** with the rare earth precursors [LnI($\eta^8$-C$_8$H$_8$)(thf)$_2$] (Ln = Y, Tb, Er) were performed, following the same procedure as before (Fig. 5a)[42].

After filtration, single crystals of the respective complexes were obtained from *n*-pentane in crystalline yields ranging from 35% to 55%, which were characterized using SCXRD and found to be isostructural to the previously synthesized stibolyl complexes (Fig. 5b). Complexes **5-Y**, **5-Tb** and **5-Er** all crystallize in the orthorhombic space group *Pnma*.

Similar to the stibolyl complexes, the central metal atoms are coordinated by the COT ring in $\eta^8$-fashion and by the bismolyl ring in $\eta^5$-fashion. The bond lengths between bismuth and the rare earth atoms vary: **5-Er** exhibited the shortest distance with 3.1502(8) Å, followed by **5-Y** with 3.1716(2) Å and **5-Tb** with 3.1879(3) Å. This is in good alignment with the increasing ionic radii from Er to Y to Tb. The slight elongation of the Bi-Ln bonds in **5-Er** and **5-Y** compared to the Sb-Ln bond lengths in their antimony counterparts can be attributed to the larger atomic radius of bismuth[47]. A similar trend is observed for the Ln-Ct$_{Bi}$ distances, with 2.374(5) Å for **5-Er**, 2.405(4) Å for **5-Y** and 2.432(2) Å for **5-Tb**. The Ln-Ct$_{COT}$ distances also show a correlation with the size of the central atom, with **5-Er** exhibiting the shortest distance of 1.701(5) Å and **5-Tb** displaying the longest distance of 1.763(2) Å. The bond angles between Ct$_{COT}$-Ln-Ct$_{Bi}$ measure at 169.4(2)° for **5-Er**, 169.63(2)° for **5-Y** and 168.594(1)° for **5-Tb**, showing a negligible influence of the size of the central atom on the linearity of the complex.

Notably, when comparing the two isostructural complexes **3-Y** and **5-Y**, the only major difference between the two structures is the

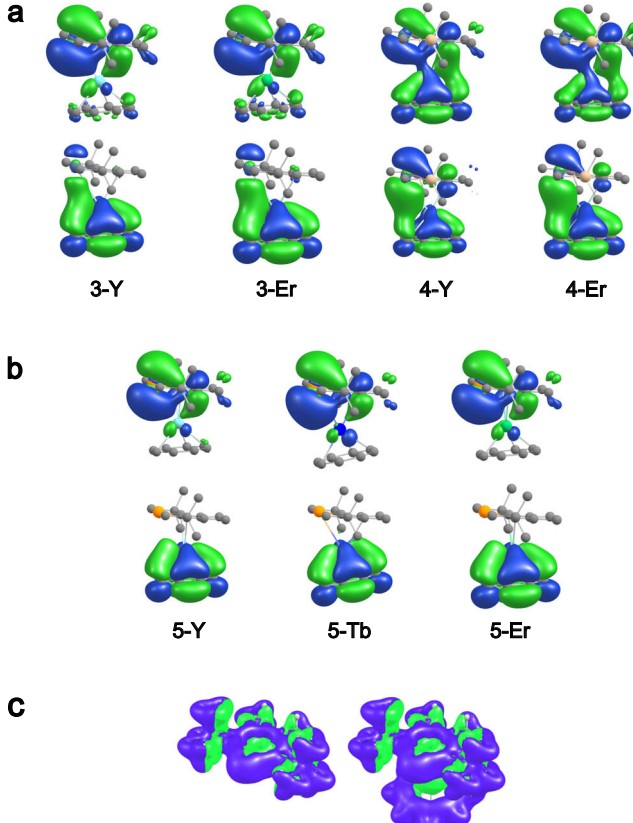

**Fig. 4 | DFT calculations (def2-TZVP/PBE0). a** HOMO (top row) and second MO (bottom row) under consideration for compounds **3** and **4**. **b** HOMO (top row) and second MO (bottom row) under consideration for compounds **5**. For open-shell compounds, the respective α orbital is depicted. Hydrogen atoms are omitted for clarity. Gray: C, beige: Si, light blue: Y, grey blue: Sb, orange: Bi, green: Er, dark blue: Tb. Note that compared to Figs. 3b and 5b the molecules are rotated counterclockwise around the CtCOT Ln-CtSb/Bi axis by 90°. **c** Current densities of the free monoanionic bismolyl ligand (left) and **5-Er** (right). Contours were drawn at ± 0.03 atomic units. Diatropic contributions are depicted in blue, paratropic contributions in green. The external magnetic field is perpendicular to the bismolyl ring plane.

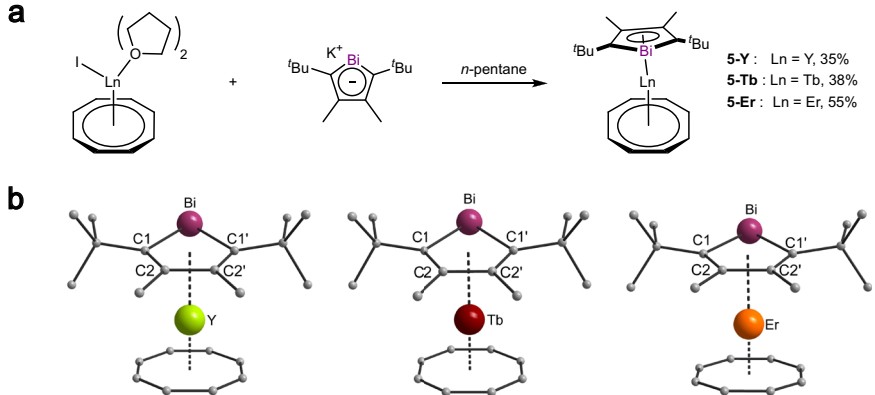

**Fig. 5 | Synthesis and Structures of rare earth bismolyl complexes. a** Synthesis of yttrium (**5-Y**), terbium (**5-Tb**) and erbium (**5-Er**) bismolyl-complexes. **b** Molecular structures of compounds **5-Y** (left), **5-Tb** (middle) and **5-Er** (right) in the solid state. For better clarity hydrogen atoms are omitted[42].

difference in the Y-Sb (3.1141(8) Å) and Y-Bi (3.1716(2) Å) bond length, reasonably in line with the increase of the covalent radius by 0.09 Å[51]. The Y-Bi bond in **5-Y** is significantly shorter than the one in [(C$_5$Me$_4$H)$_2$Y(μ-η$^1$-(BiC$_4$Me$_4$)]$_2$ (3.392(1) Å) indicating the strong difference between η$^1$ and η$^5$ coordination of the bismolyl ring[33].

The Y-Ct distances to the stibolyl and bismolyl ligands as well as the COT ligand are almost identical in both complexes. The same is true for the angle between the yttrium and the two centroids of the ligands.

Complex **5-Y** was also characterized by NMR spectroscopy, showing only subtle differences compared to **3-Y** in the $^1$H NMR spectrum. However, the most significant distinction between the two compounds was observed for the heterocyclic carbon atoms in the $^{13}$C{$^1$H} NMR spectrum. In the spectrum of **5-Y**, the signal corresponding to the α-carbons appeared at δ = 210.0 ppm, whereas the analogous antimony compound displayed a signal at δ = 187.5 ppm. The carbon atoms of the COT ligand appear as a doublet at δ = 94.6 ppm with a coupling constant of $^1J_{C,Y}$ = 2.8 Hz. In general, compounds **3-Ln, 4-Ln,** and **5-Ln** do not show decomposition in the solid state at room temperature for up to weeks and remain stable for even longer periods at -30 °C. However, particularly the bismolyl complexes decompose in solution within a few hours, accompanied by the formation of a black precipitate.

Notably, these complexes stand as a rare example of molecular rare earth-bismuth bonds, with only a limited number of instances documented thus far[33,52–54]. It is worth noting that other occurrences of rare earth-bismuth bonds often involve cluster structures[55–58].

Similar to compounds **3** and **4**, the total Mulliken overlap populations between Bi and the central rare earth metal in compounds **5** amount to approximately 0.3 electrons (see Table S5 in the Supplementary Information, again roughly half of the overlap observed for a typical single bond, e.g., 0.6 for BiH$_3$). In comparison to **4**, the HOMO contributes to a larger extent to the overlap, while the second MO considered above is decreased in relevance (see Fig. 4b and Table S5).

Again, the HOMO describes a π-bonding interaction between one of the π orbitals of the stibolyl ligand with a 4d (Y) or 5d (Tb, Er) orbital of the rare earth element, and the second relevant MO describes a δ-bonding interaction of one of the COT π orbitals with another 4d (Y) or 5d (Tb, Er) orbital of the rare earth metal. Contrary to compounds **3** and **4**, the group 15 element does not play a significant role in the latter orbital. Mulliken population analyses indicate that contributions of the Bi atom to the orbitals mostly stem from its 6p orbital, whereas contributions of the rare earth ions are dominated by their 4d (Y) or 5d (Tb, Er) orbitals (refer to Supplementary Information, Tables S6 and S7).

As congeners of the Cp ligand, the stibolyl and bismolyl anions are expected to be aromatic in nature and should retain their aromaticity upon coordination to metal ions. A clear indication of this is given by the planarity of the rings. Another indication of aromaticity, according to the magnetic criterion, is the occurrence of magnetically induced net diatropic ring currents[59–62]. Thus, ring current calculations were carried out for the complexes reported herein with the program GIMIC[63–65] using perturbed densities calculated with TURBOMOLE[66,67] (Fig. 4c, also see in the Supplementary Notes).

For the free stibolyl and bismolyl anions as well as in the complexes **3-5**, net diatropic ring currents in the range of 8-11 nA/T were found, reflecting moderate aromaticity for the bare systems, which is retained upon coordination to the rare earth ions (see Table S3 in the Supplementary Information). This is also exemplified in Fig. 4c, showing contour plots of the current density of the bare bismolyl ring and the bismolyl ligand in **5-Er**.

As documented in Table S3, the influence of the central atom, the influence of the chosen pnictogen, and the effect of replacing the $^t$Bu group with SiMe$_3$ on the calculated ring currents are minor.

Due to their large spin-orbit coupling and crystal field splitting, rare earth ions exhibit strong single-ion anisotropy. To be useful as a potential single-molecule magnet, the ligands coordinating the rare earth ion must stabilize the $m_J$ = ± $J$ state (Kramers doublet for half-integer angular momenta) and destabilize the other $m_J$ states. For Er$^{III}$ the desired ground state is the $m_J$ = ± 15/2 state. Because this state possesses a prolate electron distribution, the desired stabilization can be achieved by a strong crystal field in the equatorial plane. In the present case, the required crystal field is engendered by the relatively closely coordinating COT ligand. It is furthermore beneficial in the case of Er$^{III}$ that the $m_J$ = ± 13/2 doublet and the other $m_J$ states show a more oblate charge density which is unfavorable in an equatorial ligand field[68]. This leads to an increase of the energetic separation between the two $m_J$ states and thus to a large energy barrier[68]. However, this is a very idealized view, as in reality many of the higher states are not pure $m_J$ states but mixtures of different states (see below). In this idealized regard, the Er$^{III}$ species reported in this work possess all the needed premises for a well-performing SMM. This is confirmed by state-average complete active space self-consistent field spin-orbit (CASSCF-SO) calculations. CASSCF-SO calculations for **3-Er, 4-Er** and **5-Er** using OpenMolcas[69–71], predict a highly axial ground state in all cases (Tables S19-S21, Supplementary Information), with the anisotropy axes nearly perpendicular to the COT ligand (see Figure S80). Moreover, projection of the spin-orbit states onto a crystal field Hamiltonian employing SINGLE_ANISO[70,71] show that the ground doublet for each complex is predominantly the ±15/2 state (>99%), with g-values approaching the Ising limits, i.e., $g_x$ = $g_y$ = 0 and $g_z$ ~ 18. The first and second excited states are also found to be mostly ±13/2, while the $m_J$ composition of the remaining Kramers doublets are largely mixed. Based on the CASSCF-SO calculated electronic characteristics, the magnetic properties of the complexes described herein are expectedly highly anisotropic with barriers ~ 200 cm$^{-1}$, with relaxation occurring through the second excited state.

Hence, to probe the magnetic characteristics of the systems, static magnetic measurements were carried out. Temperature-dependent magnetic susceptibility measurements of the compounds **3-Er, 4-Er** and **5-Er** (Figures S36, S39 and S42) show high temperature $\chi T$ values of 11.26 cm$^3$ · K · mol$^{-1}$ for **4-Er**, 12.32 cm$^3$ · K · mol$^{-1}$ for **3-Er** and 10.18 cm$^3$ · K · mol$^{-1}$ for **5-Er**. The values for the stibolyl complexes **3-Er** and **4-Er** are very similar to the free ion Er$^{III}$ value of 11.48 cm$^3$ · K · mol$^{-1}$, while the bismolyl complex **5-Er** is slightly lower[72]. The lower high-temperature value of the molar susceptibility might result from the sample preparation. As the sample is fixed in an ampule with eicosane, the compounds might mix slightly resulting in a lower effective mass and thus a lower $\chi T$ value. For **4-Er** and **3-Er**, $\chi T$ as a function of temperature stays relatively constant at the high-temperature value upon cooling down to around 75 K, before slowly decreasing and then dropping fast below 11 K. The temperature susceptibility product of **5-Er** on the other hand decreases over the complete temperature range before also dropping fast below 8.5 K. These drops of $\chi T$ may indicate intermolecular antiferromagnetic coupling at low temperatures, which has been reported previously for a number of erbium compounds[22,24,48], or slow dynamics of the magnetic moment. In view of the crystal structures, intermolecular interactions appear less likely.

Isothermal magnetization measurements show a strong field dependence at low temperatures for all three compounds (Figures S37, S40 and S43). This dependence is linear at lower fields for **4-Er**, while for **3-Er** and **5-Er** it shows a sigmoidal shape, indicating a non-equilibrium state of the magnetization at these low fields, or weak intermolecular interactions. **3-Er** has the highest saturation magnetization of 5 $N_A$·μ$_B$ followed by **4-Er** with 4.6 $N_A$·μ$_B$ and **5-Er** with 4 $N_A$·μ$_B$. The saturation magnetization is reached for all compounds at 7 kOe and slightly lowers with increasing temperature up

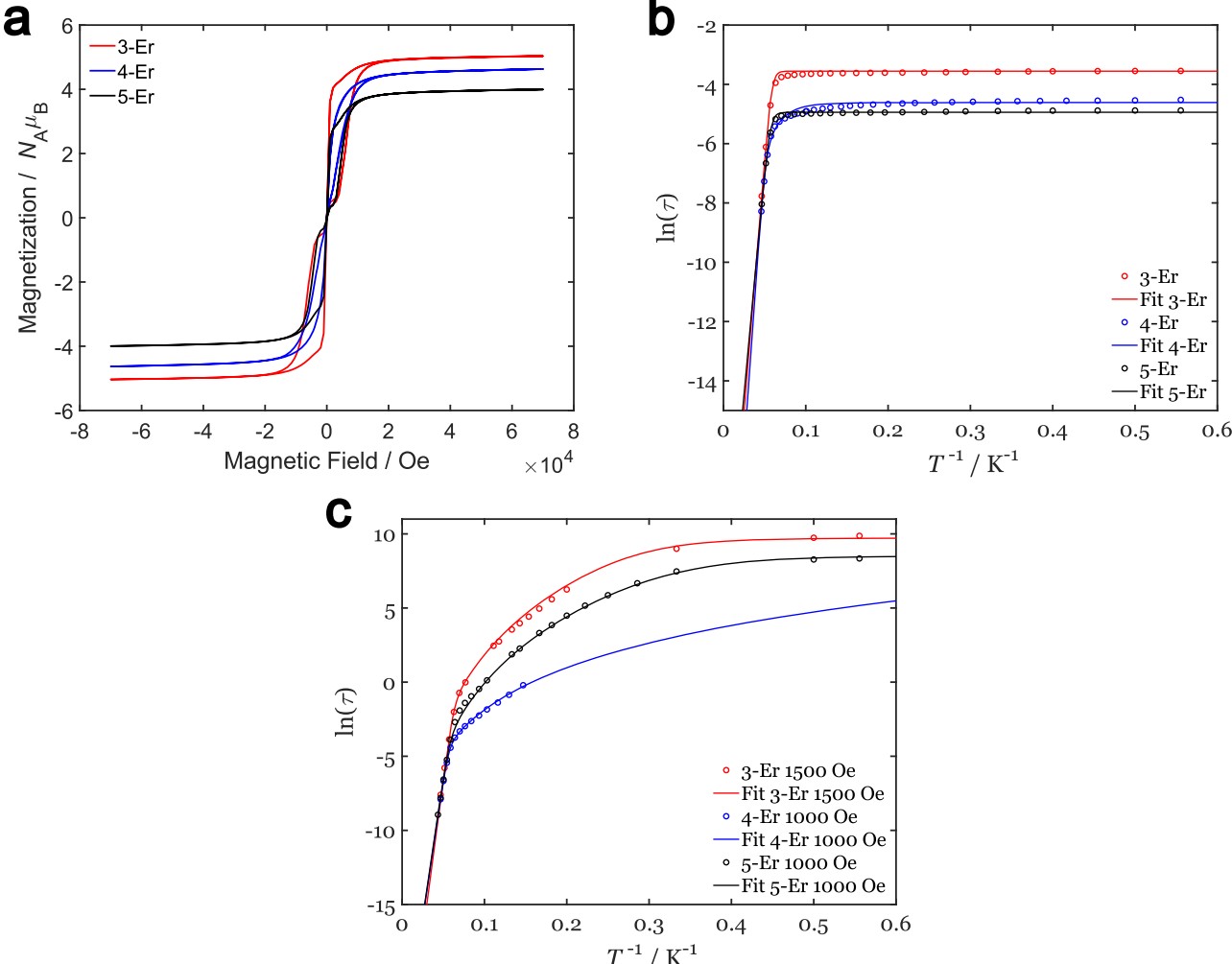

**Fig. 6 | Magnetic properties of the Erbium compounds. a** Hysteresis measurements at 1.8 K of the three erbium compounds. The magnetic field was stabilized at each measured data point. **b** Arrhenius plots of **3-Er**, **4-Er** and **5-Er**. Fit parameters are listed in Table 1. Detailed Arrhenius plots of the compounds with contributions of individual relaxation mechanisms, the fitting procedure and calculated error bars can be found in the Supplementary Notes (Figures S48, S59, and S69). **c** Arrhenius plots of the erbium compounds in applied DC Field. Fit parameters are listed in Table 2. Detailed Arrhenius plots of the compounds with broken down relaxation pathways, the fitting procedure and calculated error bars can be found in the Supplementary Notes.

to 10 K. The CASSCF calculated $\chi T(T)$ and $M(H)$ are in good agreement with the experimental data (Figures S36–S43, Supplementary Information).

Hysteresis measurements at 1.8 K (Fig. 6a), show a waist-restricted hysteresis loop, which is typical for COT $Er^{III}$ compounds, and indeed for most rare earth-based single-ion magnets[22,24,48].

The largest opening of the hysteresis loop is observed for **3-Er** with 7000 Oe and the smallest for **4-Er** with 3000 Oe. **5-Er** lies between those values with 4000 Oe. **4-Er** also has the smallest opening at non-zero field at higher temperatures, whereby the hysteresis loop is already completely closed at 5 K, while **5-Er** still shows a small open loop at non-zero field up to 7 K and for **3-Er** even up to 9 K (Figures S38, S41 and S44). This could indicate that less thermal energy is needed for the relaxation pathways of **4-Er** or be a result of the faster Raman and tunneling rate of **4-Er** (see Tables 1 and 2). The closing temperature of the hysteresis loop, in general, matches the temperature, where the zero-field cooled (ZFC) and field cooled (FC) magnetization measurements start to overlap (Figures S77-79). The trend of loop-closing temperatures of the compounds is preserved in the ZFC-FC measurements, but the values are generally lower. This could be explained by the different sweep speeds of the hysteresis compared to the ZFC-FC measurement, as both measurements are

heavily influenced by the sweep rate. Comparing the hysteresis of **3-Er** with its arsolyl congener shows very similar behavior. Both show a strong opening at non-zero fields, while the remnant magnetization rapidly diminishes at near-zero-field. Additionally, the loop opening is significantly larger for the stibolyl compound[25]. The bismolyl compound **5-Er** behaves also similarly to its congener. As with **3-Er** it exhibits a larger hysteresis opening. QTM only dominates at lower fields compared to the arsolyl. **4-Er** on the other hand shows a smaller opening of the waist-restricted hysteresis compared to its pholpholyl congener and the hysteresis loop already collapses at lower temperatures[24].

As the presence of magnetic hysteresis indicates slow relaxation of the magnetization, further alternating current (AC) magnetometry measurements were carried out for a detailed view of the relaxation behavior. In this regard, the obtained data was fitted with the generalized Debye model (equations S1 & S2)[73]. For **3-Er**, the AC data is shown in Figures S45–S47. The out-of-phase susceptibility shows a maximum with decreasing value at a constant AC frequency of around 5.6 Hz from 1.8 K to 10 K. At higher temperatures, the maximum shifts to higher frequencies up to 26.5 K, where it reaches the upper-frequency limit of the magnetometer (1 kHz). The behavior below 10 K is assigned to predominant quantum tunneling of magnetization

**Table 1 | Fit parameters of the Arrhenius fits from Fig. 6b**

| Compound | $\tau_0^{-1}$ / s$^{-1}$ | $U_{\mathrm{eff}}$ / cm$^{-1}$ | $\tau_{\mathrm{tunnel}}^{-1}$ / s$^{-1}$ | $n$ | $\tau_{\mathrm{Raman}}^{-1}$ / K$^{-n} \cdot$ s$^{-1}$ |
|---|---|---|---|---|---|
| 3-Er | $1.7 \cdot 10^{10} \pm 0.2 \cdot 10^{10}$ | $236 \pm 2$ | $35 \pm 1$ | - | - |
| 4-Er | $2 \cdot 10^{11} \pm 2 \cdot 10^{11}$ | $271 \pm 10$ | $101 \pm 4$ | $4.9 \pm 0.6$ | $2 \cdot 10^{-4} \pm 3 \cdot 10^{-3}$ |
| 5-Er | $3.6 \cdot 10^{9} \pm 0.4 \cdot 10^{9}$ | $209 \pm 2$ | $140 \pm 2$ | - | - |

**Table 2 | Fitting parameters of the in-field Arrhenius fits from Fig. 6c**

| Compound | $\tau_0^{-1}$ / s$^{-1}$ | $U_{\mathrm{eff}}$ / cm$^{-1}$ | $\tau_{\mathrm{tunnel}}^{-1}$ / s$^{-1}$ | $n$ | $\tau_{\mathrm{Raman}}^{-1}$ / K$^{-n} \cdot$ s$^{-1}$ |
|---|---|---|---|---|---|
| 3-Er | $8 \cdot 10^{11} \pm 3 \cdot 10^{11}$ | $289 \pm 5$ | $0.06 \cdot 10^{-3} \pm 8.8 \cdot 10^{-3}$ | $6.8 \pm 0.4$ | $2 \cdot 10^{-8} \pm 2 \cdot 10^{-8}$ |
| 4-Er | $1.1 \cdot 10^{11} \pm 0.12 \cdot 10^{10}$ | $261 \pm 2$ | - | $4.1 \pm 0.3$ | $5 \cdot 10^{-4} \pm 4 \cdot 10^{-4}$ |
| 5-Er | $1 \cdot 10^{11} \pm 1 \cdot 10^{11}$ | $263 \pm 22$ | $0.0002 \pm 14$ | $6.50 \pm 13.5$ | $0.0034 \cdot 10^{-5} \pm 1.3 \cdot 10^{-5}$ |

(QTM), whilst thermally activated processes become important at higher temperatures. Furthermore, the α parameter of the fit, which describes the spread of relaxation times, is constant at about α = 0.16 in the region where QTM is dominant[74]. This indicates several different QTM pathways.

The dynamic susceptibility measurements at zero applied field of **4-Er** (Figures S56–S58) and 5-Er, (Figures S66–S68) show comparable behavior to that of **3-Er**. The maximum of χ" for **4-Er** is found at 15 Hz and 1.8 K and increasing slowly to 20 Hz at 8 K. Afterwards, a strong temperature dependence of the frequency maximum is observed. For **5-Er**, the maximum of the fit at 1.8 K is at 20 Hz, slowly increasing to around 26 Hz at 10 K, before the relaxation time decreases rapidly at higher temperatures. Of note is that the temperature at which the fast decay of magnetization sets in for all three compounds maps to the temperature where the waist-restricted hysteresis completely closes. For **5-Er** the spread of relaxation time, given by the parameter α is similar to **3-Er** with values around α = 0.145 from 1.8 K to 9.5 K. A decrease of α is observed at higher temperatures (Table S15). Divergent from this, the spread of relaxation times slowly lowers for **4-Er** from α = 0.166 at 1.8 K to α = 0.145 at 8.3 K. α decreases for all compounds at higher temperatures where QTM is absent, indicating similar relaxation characteristics at higher temperatures. The temperature dependence of the relaxation rates is shown in Fig. 6b for all compounds. The data was fitted with a sum of an Orbach process and QTM (Eq. 1). Here, $\tau_0^{-1}$ is the prefactor of the Orbach process, $U_{\mathrm{eff}}$ the energy barrier of the Orbach process, $k_B$ the Boltzmann constant, $T$ the temperature, and $\tau_{\mathrm{Tunnel}}^{-1}$ the rate of the QTM. For **4-Er** the Raman process was also added, with $\tau_{\mathrm{Raman}}^{-1}$ being the rate of the Raman process and $n$ the empirical Raman exponent.

$$\tau^{-1} = \tau_0^{-1} \cdot \exp\left(\frac{-U_{\mathrm{eff}}}{k_B T}\right) + \tau_{\mathrm{Tunnel}}^{-1} + \tau_{\mathrm{Raman}}^{-1} T^n \qquad (1)$$

The fit parameters in Table 1 are in agreement with the conclusions drawn qualitatively from the behavior of χ": i.e., at low temperatures QTM is the dominant pathway, while the Orbach process is dominant at higher temperatures.

It is notable that the rate of the Orbach process of all three compounds as well as their effective energy barriers are in the same regime. This suggests that this pathway is intrinsic to the general structural motif of the compounds and only marginally affected by the different type of stibolyl or bismolyl ligands.

On the other hand, the rate of the QTM is different for all three compounds, with **3-Er** having to lowest rate and **5-Er** the highest. Thus, the rate of tunnelling is dependent on the exact ligand structure and electronic composition. While the ligands of **3-Er** are only

bent with respect to the ligand-erbium-ligand axis, the ligands of **4-Er** also show an additional tilt with respect to each other. This indicates that the ligand field of **4-Er** potentially has a stronger transverse component than **3-Er** which would favor QTM[75,76]. The Raman exponent of $4.9 \pm 0.6$ for **4-Er** is quite common for single-ion magnets and can indicate the occurrence of a multiplet with small splitting or, more likely here, the predominantly local vibrational modes governing the relaxation[77].

By applying a magnetic DC field tunnelling is suppressed[78]. Therefore, the relaxation behavior of the compounds in a static magnetic field was investigated using AC SQUID susceptometry.

Measurements were carried out in analogy to the zero-field measurements (Figures S51–S53 for **3-Er**, S62-S64 for **4-Er** and S72-S74 for **5-Er**). The resulting fit parameters are found in Tables S10, S14 and S16. For **3-Er** and **5-Er**, the spread of relaxation times described by α remains similar to the zero-field measurements. In contrast to this, α is significantly higher for **4-Er**. For longer relaxation times, DC magnetization decay measurements were performed. The DC decay curves are shown in Figures S54 for **3-Er** and S75 for **5-Er** (fit parameters in Tables S11-S12 for **3-Er** and S17-S18 for **5-Er**). As in the AC data, a distribution of relaxation times at lower temperatures (parameter β) is observed[79,80]. A combined Arrhenius plot of DC and AC data is shown in Fig. 6c.

A comparison of Tables 1 and 2 shows a similar Orbach process with and without applied field. The effective energy barriers and the rate of the Orbach process are slightly higher when applying a small magnetic field. The high-rate constant $\tau_0$ indicates that the Orbach process is the dominant relaxation pathway at higher temperatures and although the effective barriers are higher when applying a magnetic field, the rate of the Orbach process is also higher. Note that the $U_{\mathrm{eff}}$ exceeds the gap between the ground doublet and the second excited state (~ 200 cm$^{-1}$), while it approaches the full height of the barrier, as evidenced by CASSCF-SO calculations. Comparison of the CASSCF-SO obtained energy ladder and the experimental barriers indicate that relaxation in the complexes occurs via states above the second excited state.

Furthermore, QTM is largely suppressed by applying a magnetic field, but still operates at temperatures below 2 K. Between the QTM and the Orbach regime, the Raman process is now active. For **4-Er** the Raman exponent lowers slightly when applying a field from $n = 4.6 \pm 1.2$ to $n = 4.1 \pm 0.3$, which is an indication for a change in local vibrations of the compound. For **3-Er** and **5-Er** the Raman exponent lies between 6.5 and 6.8, which is lower than the expected value for Kramers ions ($n = 9$)[77]. This hints, that not only acoustic phonons are inducing relaxation but also optical phonons, which would result in a lower Raman exponent[81,82]. Remarkably, the obtained $n$ exponents are very close to the expected value of a non-Kramers doublet with $n = 7$.

When comparing to the literature, the energy barrier values of $U_{eff} = 271 \pm 10$ cm$^{-1}$ for **4-Er** in zero field are generally larger than for the germanium and phosphorous congener with an effective barrier of 120 cm$^{-1}$ and 248 cm$^{-1}$ respectively[21,24]. **3-Er** also behaves similarly to the arsenic analogue in the AC SQUID measurements in zero field, but has a higher relaxation barrier of $U_{eff} = 236 \pm 2$ cm$^{-1}$ compared to 224 cm$^{-1}$ respectively[25]. Moreover, these barriers approach the largest barriers for Er-based SMMs, highlighting the highly equatorial ligand field exerted by the combination of the COT and stibolyl and bismolyl moieties[14,24,83–86].

CASSCF-SO calculations were performed for **3-Er**, **4-Er** and **5-Er** employing the structure of the complex as experimentally determined[69–71]. The CASSCF calculated $\chi T(T)$ and $M(H)$ are in good agreement with the experimental data (Figures S36–S43, Supplementary Information), while the anisotropy axes are found to be practically perpendicular to the COT ligand (see Figure S80). Expectedly, the electronic characteristics of the three complexes are similar with highly axial $g$-values approaching the Ising limits, i.e., $g_x = g_y = 0$ and $g_z \sim 18$ (see Figure S80 and Tables S19–S22). The $m_J$ components for the lowest eight Kramers doublets for **3-Er**, **4-Er** and **5-Er** are shown in Tables S19-S21, respectively, with the ground Kramers doublets being predominantly comprised of a $m_J = \pm 15/2$ state (>99%). The strong axial character of the ground double clearly highlights the ligands' equatorial ligand field, which stabilizes the prolate electronic distribution of Er$^{III}$. Comparison between the experimental energy barrier and the electronic energy ladder obtained from CASSCF suggests that relaxation for **3-Er** and **4-Er** occurs through the top of the barrier, while for **5-Er** might occur through the fifth excited state, hence, leading to the relatively large barriers herein reported. Inspection of the average Cartesian magnetic moment probabilities obtained from CASSCF (Tables S23–S25), as a proxy for the relaxation, shows large transition probabilities for **3-Er** and **4-Er** occurring between the first excited state and 2$^{nd}$ excited state. However, probabilities of the same order are also found between the 1$^{st}$ excited state, 3$^{rd}$ and 4$^{th}$ excited state, hence, biasing the relaxation to larger excitations. Larger transitions are also observed for the 3$^{rd}$ excited state and above, thus, causing the overall relaxation to occur through the 7$^{th}$ excited state, as shown in the SQUID experiments. For **5-Er**, a similar behavior is observed with relaxation occurring via the 5$^{th}$ excited state, instead. The difference between the relaxation of stibolyl complexes and the bismolyl can be a result of the ligand field exerted by the stibolyl moiety, causing a less equatorial field, and/or higher energy intramolecular vibrations, compared to the bismolyl unit and will be the subject of further investigations.

In summary, herein the successful synthesis of potassium stibolyl and bismolyl complexes as precursors for salt elimination reactions to obtain the rare earth sandwich complexes [($\eta^5$-C$_4$R$_4$E)Ln($\eta^8$-C$_8$H$_8$)] (E = Sb, Bi; Ln$^{III}$ = Y, Er, Tb) **3-5** is reported. These compounds are examples of $\eta^5$-coordination of stibolyl and bismolyl ligands to f-elements. Quantum chemical calculations were conducted to gain insights into the bonding situation in these complexes as well as to confirm the aromaticity of the ligands. Moreover, the erbium analogues were further studied by SQUID magnetometry. Waist-restricted hysteresis loops were observed for all systems with the relaxation dynamics at zero field dominated by the Orbach process at high temperatures and QTM at low temperatures. In all cases, the $U_{eff}$ approaches one of the largest reported barriers for Er-based SMMs, with the largest barrier occurring for the shortest stibolyl-Er distance. Our results show that mixed COT, stibolyl- and/or bismolyl- ligand variations can be further explored for the synthesis of Er-SMMs in attempts to suppress QTM at zero field and/or to obtain large barriers and blocking temperatures for Er-based SMMs.

## Methods

### Synthetic method

Experiments were carried out under a dry, oxygen-free argon atmosphere using Schlenk-line and glovebox techniques. All solvents and reagents were rigorously dried and deoxygenated before use. All compounds were characterized by single-crystal X-ray diffraction studies. Further details are available in the section Supplementary Methods.

## Data availability

All synthetic protocols, spectroscopic data (NMR, IR), Supplementary Figs. and Tables, and detailed crystallographic information, details on the quantum chemical calculations, magnetic measurements, and CASSCF calculations can be found in the Supplementary Information. All additional data are available from the corresponding author upon request. Crystallographic data for the structures reported in this Article have been deposited at the Cambridge Crystallographic Data Centre, under deposition numbers CCDC 2358764 (**3-Y**), 2358765 (**3-Er**), 2381646 (**3-Tb**), 2358766 (**4-Y**), 2358767 (**4-Er**), 2381647 (**4-Tb**), 2358768 (**5-Y**), 2358769 (**5-Tb**) and 2358770 (**5-Er**). Copies of the data can be obtained free of charge via https://www.ccdc.cam.ac.uk/structures/. Cartesian coordinates for the optimized structures are available in the Source Data file accompanying this manuscript. Source data are provided with this paper.

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

## Acknowledgements

Prof. Dr. Sarkar, Marie Leimkühler and Tabea Pfister of the Institute of Inorganic Chemistry, University of Stuttgart are thanked for preliminary magnetic measurements. SG acknowledges the Fonds der Chemischen Industrie for a Kekulé fellowship (No. 110160). E.M.-P. thanks the Alexander von Humboldt fellowship for experienced researchers for support. F.B. and F.W. gratefully acknowledge support from the Deutsche Forschungsgemeinschaft (DFG, German Research Foundation) through the Collaborative Research Centre "4f for Future" (CRC 1573, project no. 471424360, project Q). This work is supported by the German Federal Ministry of Education and Research (BMBF) under Contracts 02NUK059F. Deutsche Forschungsgemeinschaft (DFG) is acknowledged for financial support within the Reinhart Koselleck-Projekt 440644676, RO 2008/19-1.

## Author contributions

Experimental work was carried out by N.S. Magnetic measurements were carried out by V.B. Density functional theory calculations were performed by F.B., S.G. and F.W. CASSCF calculations were carried out by E.M.P. Single-crystal X-ray diffraction experiments and refinement were done by X.S. Project administration was done by P.W.R. Supervision was the responsibility of J.V.S., F.W. and P.W.R. The original draft was written by N.S., F.B., V.B., and E.M.P., with all authors contributing to review and editing.

## Funding

## Competing interests

The authors declare no competing interests.
