## [Transparent Peer Review file · Nature Communications]

Rare Earth Stibolyl and Bismolyl Sandwich Complexes

Corresponding Author: Professor Peter Roesky

Version 0:

Reviewer comments:

Reviewer #1

(Remarks to the Author)

The manuscript shows the first examples of an f-element (Er) bonding/interacting with the largely unstudied stibolyl & bismolyl complexes; the synthesis, structural characterisation, magnetic properties are presented and discussed. There is also a considerable computational study that aims to rationalise the type of bonding present in the complexes along with explaining their magnetic observations. The work is original and would be of interest to chemists in general. The methodology of how the authors have gone about the work is sound and there is sufficient detail provided so that one would be able to reproduce their work. I do have the following comments/suggestions, one of which I feel needs accounting for/explaining in greater detail:

Minor issues:

P3, Line 65: have been instead of were?

Figure 1: Please space out the complexes as they are too close together. The heteroatoms are also difficult to see due to the size of the 5-membered rings compared with the size of the element/letter, I suggest resizing these.

P7, lines 144, 147 & 149: No mention of errors for the reported phospholyl system.

Scheme 3: This has the same heteroatom issue as figure 1, please correct.

Figure 6: α is used but in text is written alpha.

P12, line 251: just have a comma to include the short sentence as part of the previous, as it stands it reads as not having a noun in the sentence.

P13, line 267: tBu and SiMe₃ are not sub/sup scripted.

P16, line 321: raman should be Raman.

P15, line 313: Waist-restricted hysteresis loop, I'm not familiar with that term... closed loop?

Figure 8: Change the term "done" to "performed" or something else, I think the supp info has a better statement.

P17, line 347: use of alpha and α in text, please pick and stick with one.

P17, line 349: use of comma is needed as again, sentence does not have a noun.

P20, line 404 & P21, line 427: Kramer vs Kramers.

Major concerns:

P13, lines 269-279: So there is mention of the GS mJ state being stabilized by the COT ligand being in an equatorial plane, yes, but then the next statement is that the first excited state is essentially higher in energy as the equatorial plane promotes the prolate state to be at a higher energy, yes, this gives a large energy barrier... but what about all of the other states, I think this needs re-writing as it is misleading.

P13, lines 279-379: Your CASSAF-SO calculations indicate a highly axial GS that is perpendicular to the COT ligand (S73 does not show the easy axis as I would expect to have been provided, I believe it should be included/shown in a figure for clarity) but that once you get to the 2nd excited state you have a large degree of mixing and this is where one would expect a transition/ U_{eff} to be... your magnetic data does not show this and indeed seems to suggest that you either get to, or exceed, the 7th excited state in 2 cases (5-Er seems to go via the 5th excited state). I agree that this is rare to see (7th excited state) but why are you seeing this? Why does one of the complexes appear to not get to the 7th excited state but the other 2 do (you even mention in text they all share the same motif and it is just Raman/vibrations being different for that relaxation pathway)? There needs to be a discussion or at least some suggestions as to why this is the case.

Reviewer #2

(Remarks to the Author)

The synthesis of two new stibolyl and bismolyl ligands are described. Previously, the tetramethyl substituted bismolyl was known, but in this case, Roesky and co-workers prepare the 1,4-bis(trimethylsilyl)-2,3-dimethyl as well as the 2,4-di-tert-butyl version for Sb, while only the di-tert-butyl bismolyl is reported.

Bringing me to question 1 - why not the bismolyl with the trimethylsilyl substitution?

The potassium salts of the stibolyl and bismolyl are reacted with (COT)Ln(I)(THF) to yield the unsolvated (COT)Ln(heterolyl). For Sb, both the Y and Er compounds are reported while Tb, Y, and Er are reported for Bi.

Question 2 - why not Tb with the stibolyl?

These are the first examples of ada-5 Sb and Bi-based heterolyls with the lanthanides. Each complex was spectroscopically and structurally characterized, and the Er compounds were found to have interesting magnetic properties. Quantum calculations were also performed to examine the bonding situation in the Ln-Sb and Ln-Bi interactions.

Overall, this shows the novel synthesis of heterolyls whose coordination chemistry have exploded in the past decade, but has lagged with the f elements, particularly with the heavier pnictogens. This is a nice contribution to the field.

Can the authors comment on the thermal stability of these molecules? Is decomposition observed quickly or are these compounds stable in solid-state/solution for some amount of time?

The heterolyl ring is planar, but what I found unique about these structures which is not evident from the figures is the tilt in the Cp. For example, the Ln-C(COT) distances are all about the same but there are two shorter Ln-C(heterolyl) bonds, two longer distances and then the long Ln-Sb or Ln-Bi distances. The authors comment on the Ln-centroid but can they look at how these Ln-C(heterolyl) bonds compare to Cp* or other cyclopentadienyl ligands?

In the introduction, it is mentioned that Demir's Y-Bi complex does not have any pi-bonding. Do these compounds? The description of the bonding needs more details. For example, the authors state, "Again, Mulliken population analyses indicate that the Bi atom mainly contributes with a p-type orbital, whereas contributions of the central rare-earth ions are dominated by a d-type orbital (see Tables S6 and S7)." OK, but what does that mean? I am a lowly synthetic chemist with a very small brain not trained in the complexities of high order calculations so help me out. Is the p-type orbital and d-type orbital forming a sigma interaction or a pi interaction or a little of both? I am not sure what p-type and d-type mean either? Do you mean a hybrid orbital with mostly p or d character?

Reviewer #3

(Remarks to the Author)

Roesky, Weigend, van Slageren, and co-workers report the synthesis and characterisation of stibolyl and bismolyl ligand-based f-element sandwich complexes. The use of a bulky di-anionic COT ligand allows them to study the dynamic magnetic behaviour of the Er(III) versions. The introduction of heavier main group elements in a cyclopentadienyl ring (CpE, E = main group element) is a recent-trend to improve single-molecule magnet (SMM) performance of metallocene-based single-molecule magnets. The idea of exploiting bismuth's strong spin-orbit coupling to influence the relaxation-behaviour of an SMM is interesting, but it will be more relevant to the systems where ligand field is arises purely or predominantly from the CpE (E = Sb, Bi) ligand.

The current work can be visualised as a continuation of their recent work on arsolyl sandwich complexes (ref 24). The magnetic behaviour 3-Er and 5-Er reveals an improved energy barrier compared to iso-structural arsenic analogue, mainly due to their longer Er-CpE(cent) distances. However, structural data suggests that in all the three Er complexes, COT sits much closer to the Er(III) center than CpE, thus the dominant ligand-field is arising from the di-anionic COT ligand. These principles are well-established in the literature. The relaxation dynamics/Orbach process of this complexes is not affected in a material sense by the different type of stibolyl and bismolyl ligands.

In the introduction, the authors mentioned that they aimed to investigate the influence of heavier atoms on metallocene-based SMMs. In that case, the authors should target using hetero-atom-based cyclopentadienyl ligands to prepare complexes of such as [(CpE)₂Ln]⁺ (E = Si, Sb and Ln(III) = Dy, Tb) so that they can fully exploit the axial ligand field exerted by these ligands to stabilize oblate like m_J states of Dy.

Besides, I have the following comments for the authors:

- (1) Why have the authors prepared the Tb(III) analogue only with the bismolyl ligand and not with the stibolyl ligand?
- (2) I was also wondering about the purpose of involving a paramagnetic metal like Tb(III) in the manuscript if you are not showing its magnetic behaviour.
- (3) Comparing the hysteresis behaviour of 3-Er, 4-Er, and 5-Er with reported P and As analogues [(COT)Er(Dtp/Dsas)] would be useful.
- (4) A complementary plot of ZFC and FC magnetic susceptibility vs temperature (for 3-Er, 4-Er, and 5-Er) could be useful to determine the hysteresis temperature.
- (5) The author's did not explain why the high-temperature chiT (cT) value of 5-Er is 11% lower than the free ion Er(III) value, and the same goes for the saturation of magnetisation value. Also, scaling with a factor of 0.85 is a bit too high and indicates possible impurities in the sample (Fig S38).
- (6) The chiT (cT) plot of 3-Er and 4-Er (Fig S32 and Fig S35) shows an unusual drop around 45 K. The authors should clarify this.
- (7) In the M vs H plot for 3-Er (Figure S33), 4-Er (Figure S36), and 5-Er (Fig S39), the data points extracted from the hysteresis measurement (low-temperature) do not match with the calculated value at low field. Therefore, experimentally, authors are not probing the same ground state as mentioned in the calculation. The authors should explain this.
- (8) The value of alpha parameter at 23.9 K and 26.5 K (Table S9), 23.3 K and 24.9 K (Table S13), and 23.9 K and 26.5 K (Table S15) is meaningless, as no maxima were observed in the corresponding X'' vs frequency plot at these temperatures (Fig S42, Fig S53, and Fig S63). Therefore, these values should be removed from the respective tables, and these

temperatures should not be included in the Arrhenius plot for estimating the energy barrier. This will probably change the energy barriers.

(9) A uniform notation of Avogadro's constant (NA) should be used throughout the manuscript (modifications needed in line 309 and 310).

(10) A uniform abbreviation of cyclooctatetraenyl ligands (COT) should be maintained throughout the text (changes required in line 446).

Thus, I think the paper is not worthy of being published in a reputed journal like Nature Communications. There is some nice synthetic chemistry but, ultimately, the performance of the SMMs is only an incremental advance based on what is already known about such systems.

Reviewer #4

(Remarks to the Author)

I co-reviewed this manuscript with one of the reviewers who provided the listed reports. This is part of the Nature Communications initiative to facilitate training in peer review and to provide appropriate recognition for Early Career Researchers who co-review manuscripts."

Version 1:

Reviewer comments:

Reviewer #1

(Remarks to the Author)

I thank the authors for making the changes that I suggested. I believe the authors have also addressed all of my major concerns that I had with the manuscript and believe it should now be published. I look forward to reading the follow up manuscripts and other works they produce.

Reviewer #2

(Remarks to the Author)

It appears that the authors have addressed the issues that I raised as well as the other reviewers satisfactorily. This is a nice contribution, and I support its publication!

Reviewer #3

(Remarks to the Author)

The authors have responded positively to most of the comments raised by the reviewers. However, this reviewer remains ambivalent as to the novelty and significance of the results. Whilst there is some interest in exploring the impact of heavier group 15 elements on the dynamic magnetic properties of the corresponding lanthanide sandwich compounds, the outcomes are broadly similar to what has previously been determined with other heavier main group metallole complexes bound to the {Er(COT)} building block. As such, this paper does not provide a major conceptual advance, but rather represents a natural extension of previous work in the field. The SMM properties of the erbium- stibolyl and bismolyl complexes are essentially the same as other Er-COT sandwiches. As such, the work is not quite at the level required for NComms and publication is not recommended.

In addition, there seems to be great reluctance on the part of the authors to provide parallel characterization of the magnetic properties of the terbium compounds. This is disappointing, since even 'negative' results will be of some use to people working in this field since they would deepen our understanding of Tb(III) in a metallocene-like environment.

Reviewer 1:

I do have the following comments/suggestions, one of which I feel needs accounting for/explaining in greater detail:

Minor issues:

P3, Line 65: have been instead of were?

We have adjusted the sentence accordingly.

Figure 1: Please space out the complexes as they are too close together. The heteroatoms are also difficult to see due to the size of the 5-membered rings compared with the size of the element/letter, I suggest resizing these.

We thank the reviewer for the comment. We have spaced out the complexes in Figure 1 as suggested and made adjustments to enhance the visibility of the heteroatoms. Additionally, we have updated Schemes 1, 2, and 3 to improve visibility and ensure consistency throughout the manuscript.

P7, lines 144, 147 & 149: No mention of errors for the reported phospholyl system.

The respective errors have been added. Please note that for the Y-COT(centroid) distance two values are given, since the COT is disordered over two positions.

Scheme 3: This has the same heteroatom issue as figure 1, please correct.

As stated above, we have adjusted this scheme along with the other schemes to improve visibility.

Figure 6: α is used but in text is written alpha.

We have changed the notation to consistently use " α " throughout the text and figures.

P12, line 251: just have a comma to include the short sentence as part of the previous, as it stands it reads as not having a noun in the sentence.

We have revised the sentence to include a comma for better readability.

Major concerns:

P13, line 267: tBu and SiMe₃ are not sub/sup scripted.

This has been fixed accordingly.

P16, line 321: raman should be Raman.

We have capitalized Raman.

P15, line 313: Waist-restricted hysteresis loop, I'm not familiar with that term... closed loop?

A waist-restricted hysteresis is generally known as a hysteresis that is open at applied magnetic field but closes as soon as no magnetic field is applied. This term is common in literature (10.1021/jacs.3c12427, 10.1038/s41467-017-01553-w).

Figure 8: Change the term "done" to "performed" or something else, I think the supp info has a better statement.

We changed it to match the Supporting Information: "The magnetic field was stabilized at each measured data point."

P17, line 347: use of alpha and α in text, please pick and stick with one.

As already stated above, we have standardized the use of ' α ' throughout the text.

P17, line 349: use of comma is needed as again, sentence does not have a noun.

We have revised the sentence to include a comma.

P20, line 404 & P21, line 427: Kramer vs Kramers.

We changed line 427 to 'Kramers' for consistency.

P13, lines 269-279: So there is mention of the GS mJ state being stabilized by the COT ligand being in an equatorial plane, yes, but then the next statement is that the first excited state is essentially higher in energy as the equatorial plane promotes the prolate state to be at a higher energy, yes, this gives a large energy barrier... but what about all of the other states, I think this needs re-writing as it is misleading.

We thank the reviewer for pointing out this simplification. We have clarified in the manuscript that the other mJ states also show more oblate charge densities.

P13, lines 279-379: Your CASSAF-SO calculations indicate a highly axial GS that is perpendicular to the COT ligand (S73 does not show the easy axis as I would expect to have been provided, I believe it should be included/shown in a figure for clarity) but that once you get to the 2nd excited state you have a large degree of mixing and this is where one would expect a transition/ U_{eff} to be... your magnetic data does not show this and indeed seems to suggest that you either get to, or exceed, the 7th excited state in 2 cases (5-Er seems to go via the 5th excited state). I agree that this is rare to see (7th excited state) but why are you seeing this? Why does one of the complexes appear to not get to the 7th excited state but the other 2 do (you even mention in text they all share the same motif and it is just Raman/vibrations being different for that relaxation pathway)? There needs to be a discussion or at least some suggestions as to why this is the case.

We thank the referee for the comments. The arrows in Figure S80 show the easy axes of the magnetization for all three complexes (green arrow). Regarding the barrier and the relaxation pathway, the referee is correct, and relaxation seems to occur through the 7th excited state for **3-Er** and **4-Er**, whilst it occurs through the 5th excited state for **5-Er**.

To rationalize this behavior, we inspect the average cartesian magnetic moment transition probabilities acting as a proxy for the relaxation. For **3-Er**, large transition probabilities are found between the ground state, the 1st excited and the second excited state. It is also observed that the transition to the same state but with opposite orientation (QTM transitions) is solely large for the 2nd excited state, hence, it is expected that relaxation occurs through the 2nd excited state. However, probabilities of the same order are also found between the 1st excited state, 3rd and 4th excited state, hence, biasing the relaxation to larger excitations. Larger transitions are also observed for the 3rd excited state and above, thus, causing the overall relaxation to occur through the 7th excited state, as can be inferred from experiments. A similar situation is also found for **4-Er**, with large transition probabilities occurring between the 1st excited state and the 2nd and 5th excited state, biasing relaxation through higher levels. In **5-Er** we find a similar situation, however, above the 5th excited state, the transition probabilities decrease, hence, prompting relaxation through this state.

A possibility of such behavior could be a larger axial ligand field exerted by the Sb-ligands compared to the Bi. This can also be observed by the closest contact between the Cp ligand and the Dy ion for the Sb and SbBi system compared to the Bi one. Other possibilities also include larger vibrational energies which decrease the occurrence of phonon-mediated relaxation and will be the subject of further studies.

Reviewer 2:

The synthesis of two new stibolyl and bismolyl ligands are described. Previously, the tetramethyl substituted bismolyl was known, but in this case, Roesky and co-workers prepare the 1,4-bis(trimethylsilyl)-2,3-dimethyl as well as the 2,4-di-tert-butyl version for Sb, while only the di-tert-butyl bismolyl is reported.

Bringing me to question 1 - why not the bismolyl with the trimethylsilyl substitution?

We have also prepared the potassium bismolyl with trimethylsilyl substitution, however the yields were lower compared to the other complexes. Additionally, the crystallization of bismolyl complexes seemed to work drastically better for the t-Bu substituted bismolyl. Since we used only single crystalline material for all analytics including the magnetic measurements, we did not proceed in synthesizing the bismolyl ligand with the trimethylsilyl substitution. We also added one sentence stating this to the manuscript: 'Due to difficulties in preparation, of the ligand and the corresponding metal complexes, we did not synthesize the Bi analogue to 2-Sb'.

Question 2 - why not Tb with the stibolyl?

The Tb bismolyl complex was synthesized to examine possible luminescence properties, as Tb(COT)I(thf)₂ displays intense emission after excitation under UV light. Additionally, bismuth heterocycles have been reported to show phosphorescence (<https://doi.org/10.1021/acs.inorgchem.8b00149>). Despite this, our complex did not display luminescence, which is why we stopped exploring the terbium compounds. However, we have now synthesized the corresponding terbium complexes and included them in Scheme 2 as well as added them to the supporting information.

Can the authors comment on the thermal stability of these molecules? Is decomposition observed quickly or are these compounds stable in solid-state/solution for some amount of time?

Our findings indicate that the compounds are relatively stable in the solid state at room temperature for up to weeks and remain stable for longer periods at -30°C. However, when measuring NMR in C₆D₆, we observe that particularly the bismolyl complexes decompose in solution after a few hours, accompanied by the formation of a black precipitate. Therefore, all samples were always stored at -30°C. The following sentences were added in the manuscript on page 8, to address this point: 'In general, compounds 3-Ln, 4-Ln, and 5-Ln do not show decomposition in the solid state at room temperature for up to weeks and remain stable for even longer periods at -30°C. However, particularly the bismolyl complexes decompose in solution within a few hours, accompanied by the formation of a black precipitate.'

The heterolyl ring is planar, but what I found unique about these structures which is not evident from the figures is the tilt in the Cp. For example, the Ln-C(COT) distances are all about the same but there are two shorter Ln-C(heterolyl) bonds, two longer distances and then the long Ln-Sb or Ln-Bi distances. The authors comment on the Ln-centroid but can they look at how these Ln-C(heterolyl) bonds compare to Cp* or other cyclopentadienyl ligands?

We thank the referee for this remark. Indeed, the heterolyl rings are slightly tilted, resulting in two longer and two shorter Ln-C bond lengths, which is also found in related phospholyl (<https://doi.org/10.1039/C8SC01626G>) and arsolyl (<https://doi.org/10.1021/acs.inorgchem.3c03374>) complexes.

The following sentence containing information about the tilting as well as a comparison with two Cp ligands (Cp* and the more sterically demanding Cp^{†††}) was added to the manuscript: 'In all complexes, the stibolyl ring is notably tilted due to the steric demands of the antimony atom and the ^tBu groups, resulting in Er-C1 and Er-C2 bond lengths in **3-Er** of 2.757(3) and 2.650(3) Å, respectively. This tilting is absent in [Er(COT)Cp*] (Er-C distance of 2.575(3)-2.579(3) Å).⁴⁶ However, in the more sterically demanding [Er(COT)Cp^{†††}] (Cp^{†††} = C[†]tBu₃H₂), the Er-C distances range from 2.580(2) to 2.649(2) Å, demonstrating a slight tilt of the Cp ligand, however not quite as pronounced as in our stibolyl complexes.⁴⁷

In the introduction, it is mentioned that Demir's Y-Bi complex does not have any pi-bonding. Do these compounds? The description of the bonding needs more details. For example, the authors state, "Again, Mulliken population analyses indicate that the Bi atom mainly contributes with a p-type orbital, whereas contributions of the central rare-earth ions are dominated by a d-type orbital (see Tables S6 and S7)." OK, but what does that mean? I am a lowly synthetic chemist with a very small brain not trained in the complexities of high order calculations so help me out. Is the p-type orbital and d-type orbital forming a sigma interaction or a pi interaction or a little of both? I am not sure what p-type and d-type mean either? Do you mean a hybrid orbital with mostly p or d character?

We thank the referee for pointing out the difficulties associated with the interpretation of the quantum chemical results. We have addressed this in the manuscript by discussing the bonding situation within the complexes in more detail.

Reviewer 3:

(1) Why have the authors prepared the Tb(III) analogue only with the bismolyll ligand and not with the stibolyll ligand?

The Tb bismolyll complex was synthesized to examine possible luminescence properties, as Tb(COT)l(thf)₂ displays intense emission after excitation under UV light. Additionally, bismuth heterocycles have been reported to show phosphorescence (<https://doi.org/10.1021/acs.inorgchem.8b00149>). Despite this, our complex did not display luminescence, which is why we stopped exploring the terbium compounds. However, we have now synthesized the corresponding terbium complexes and included them in Scheme 2 as well as added them to the supporting information.

(2) I was also wondering about the purpose of involving a paramagnetic metal like Tb(III) in the manuscript if you are not showing its magnetic behaviour.

As mentioned in the previous answer, we hoped for interesting photophysical properties employing Terbium.

(3) Comparing the hysteresis behaviour of 3-Er, 4-Er, and 5-Er with reported P and As analogues [(COT)Er(Dtp/Dsas)] would be useful.

We have added a comparison to the respective congeners in the manuscript.

(4) A complementary plot of ZFC and FC magnetic susceptibility vs temperature (for 3-Er, 4-Er, and 5-Er) could be useful to determine the hysteresis temperature.

We thank the reviewer for pointing this out. We have added the measurements to the ESI (figures S77-79) and added a short paragraph in the main body comparing the measurements.

(5) The author's did not explain why the high-temperature χT (cT) value of 5-Er is 11% lower than the free ion Er(III) value, and the same goes for the saturation of magnetisation value. Also, scaling with a factor of 0.85 is a bit too high and indicates possible impurities in the sample (Fig S38).

We thank the referee for the comments. This deviation might be explained by the large crystal field splitting of the ground multiplet, which is in a similar energy regime as the thermal energy at room temperature. Thus, not all energy levels are equally occupied and the χT value differs from the free ion value. As the magnetization vs. applied field measurements at low temperatures only probes the lowest doublet of the erbium molecule because of the crystal field splitting, there is nothing inherently connecting it to the magnetization value of the free ions. As for the calculated values: Since they are calculated with the molecule in the gas phase, they do not consider e.g. intermolecular interactions or other packing effects and thus the calculation might be off by a small amount.

(6) The χT (cT) plot of 3-Er and 4-Er (Fig S32 and Fig S35) shows an unusual drop around 45 K. The authors should clarify this.

We thank the referee for the comment. This drop might result from a crystal phase change at this temperature, e.g. a small shift in the cp or COT ligand, which would change the crystal field a small amount and thus influence the magnetic properties. Similar instabilities were also observed with the arsoly and less pronounced for the phospholy congener (10.1021/acs.inorgchem.3c03374, 10.1039/C8SC01626G).

(7) In the M vs H plot for 3-Er (Figure S33), 4-Er (Figure S36), and 5-Er (Fig S39), the data points extracted from the hysteresis measurement (low-temperature) do not match with the calculated value at low field. Therefore, experimentally, authors are not probing the same ground state as mentioned in the calculation. The authors should explain this.

The referee is correct, and the experimental M(H) loops do not fully match with the ones obtained from CASSCF M(H) results, especially at low field/temperature. However, this is not unexpected at all. Commonly, although CASSCF is very powerful for the determination of the electronic characteristics of lanthanide-based SMMs, these ab initio calculations do not include several other effects such as intermolecular interaction, distortions in the crystal, solvent effect, etc., which can cause small deviations of the electronic characteristics of the systems, often visible at low temperature/fields. Moreover, the highly anisotropic nature of lanthanide-SMMs makes the systems depend strongly on sample orientation and polycrystalline effects, which might be visible in the low temperature/low field region. Considering the limitations of CASSCF and the CASSCF obtained M(H) traces, the nature of the sample and its highly air-sensitive characteristics, we find the theoretical and experimental results to be in good agreement in all cases.

(8) The value of alpha parameter at 23.9 K and 26.5 K (Table S9), 23.3 K and 24.9 K (Table S13), and 23.9 K and 26.5 K (Table S15) is meaningless, as no maxima were observed in the corresponding X'' vs frequency plot at these temperatures (Fig S42, Fig S53, and Fig S63). Therefore, these values should be removed from the respective tables, and these temperatures should not be included in the Arrhenius plot for estimating the energy barrier. This will probably change the energy barriers.

We thank the referee for the comments. We have removed the respective values from the figures and the tables and updated the Arrhenius fits. The removal of the high temperature values did indeed lower the effective energy barrier by a small amount.

(9) A uniform notation of Avogadro's constant (N_A) should be used throughout the manuscript (modifications needed in line 309 and 310).

We have changed ' N_a ' to ' N_A ' to match the rest of the manuscript.

(10) A uniform abbreviation of cyclooctatetraenyl ligands (COT) should be maintained throughout the text (changes required in line 446).

We have changed 'Cot' to 'COT'.